# Dynamic Changes in the Microbial Composition and Spoilage Characteristics of Refrigerated Large Yellow Croaker (*Larimichthys crocea*) during Storage

**DOI:** 10.3390/foods12213994

**Published:** 2023-10-31

**Authors:** Binbin Li, Shuji Liu, Xiaoting Chen, Yongchang Su, Nan Pan, Dengyuan Liao, Kun Qiao, Yihui Chen, Zhiyu Liu

**Affiliations:** 1Institute of Postharvest Technology of Agricultural Products, College of Food Science, Fujian Agriculture and Forestry University, Fuzhou 350002, China; bbli3606@163.com; 2Key Laboratory of Cultivation and High-Value Utilization of Marine Organisms in Fujian Province, National Research and Development Center for Marine Fish Processing (Xiamen), Fisheries Research Institute of Fujian, Xiamen 361013, China; xtchen@jmu.edu.cn (X.C.); suyongchang@stu.hqu.edu.cn (Y.S.); npan01@qub.ac.uk (N.P.); liaodengyuan@sina.com (D.L.); qiaokun@xmu.edu.cn (K.Q.); 13906008638@163.com (Z.L.)

**Keywords:** diversity, 16S rDNA sequencing, dominant spoilage bacteria, *Pseudomonas*

## Abstract

The quality changes, dynamic changes in microbial composition, and diversity changes in large yellow croaker *(Larimichthys crocea)* during 4 °C refrigeration were studied using 16S rDNA high-throughput sequencing technology, and the total viable count (TVC), total volatile basic nitrogen (TVB-N), and thiobarbituric acid-reactive substances (TBARS) were determined. The results revealed a consistent increase in TVC, TVB-N, and TBARS levels over time. On the 9th day, TVC reached 7.43 lg/(CFU/g), while on the 15th day, TVB-N exceeded the upper limit for acceptable quality, reaching 42.56 mg/100 g. Based on the 16S rDNA sequencing results, we categorized the storage period into three phases: early storage (0th and 3rd days), middle storage (6th day), and late storage (9th, 12th, and 15th days). As the storage time increased, both the species richness and diversity exhibited a declining trend. The dominant genus identified among the spoilage bacteria in refrigerated large yellow croaker was *Pseudomonas*, accounting for a high relative abundance of 82.33%. A comparison was carried out of the spoilage-causing ability of three strains of *Pseudomonas* screened and isolated from the fish at the end of storage, and they were ranked as follows, from strongest to weakest: *P. fluorescen*, *P. lundensis*, and *P. psychrophila*. This study will provide a theoretical basis for extending the shelf life of large yellow croaker.

## 1. Introduction

Large yellow croaker (*Larimichthys crocea*) is a significant commercially farmed marine fish species in China, highly favored by consumers due to its abundance of high-quality protein and polyunsaturated fatty acids [1]. However, due to its high moisture content, large yellow croaker is prone to decay and deterioration caused by protein degradation, lipid oxidation, and microbial activity after death. Among these factors, microbial growth and metabolism play a decisive role [2]. Studies have revealed that while the fish’s body initially harbors a variety of microorganisms, with prolonged storage time, only a portion of these bacteria gain dominance and actively participate in the process of decay [3]. These specific spoilage bacteria degrade nitrogen-containing compounds in the fish body, resulting in the production of trimethylamine, ammonia, alcohols, ketones, aldehydes, and other metabolites at low levels, ultimately leading to the deterioration of quality and spoilage [4]. Therefore, investigating the quality changes and dynamic changes in microbial composition during the storage process is helpful for providing a theoretical basis for studying preservation techniques for large yellow croaker.

High-throughput sequencing technology, also referred to as “next-generation sequencing technology”, is a cultivation-independent microbial research technique that offers advantages such as high throughput, fast speed, and exceptional accuracy. High-throughput sequencing has been extensively employed in the investigation of microbial communities across diverse fields, including food [5,6,7,8], the environment [9,10], and medicine [11,12]. Currently, Illumina sequencing [13,14] stands out as the most widely utilized method. In recent years, 16S rDNA has gained significant prominence for studying microbial composition changes and microbial diversity due to its large information content within the molecular sequence, along with its stability and ease of extraction and analysis. Syropoulou et al. [15] utilized high-throughput sequencing technology to analyze changes in microbial composition on whole European sea basses as well as fish fillets under aerobic- and modified-atmosphere packaging conditions at different storage temperatures. The study revealed the presence of *Pseudomonas*, *Acinetobacter*, and *Psychrobacter* genera in both the initial fish and fillets; however, as the storage time extended, the relative abundance of *Pseudomonas* increased significantly, and it eventually became the dominant genus. Furthermore, a combination of high-throughput sequencing and other techniques was employed to analyze the relationship between microbial composition and changes in product quality during storage. Sun et al. [16] conducted a joint analysis using headspace gas chromatography–ion mobility spectrometry and high-throughput sequencing. They discovered a strong positive correlation between the production of flavor compounds such as alcohols, acids, esters, and phenols and the presence of the *Mitsuokella* genus in kimchi. They also observed that the *Lactobacillus* genus and the *Weissella* genus were strongly associated with the production of aldehydes and ketones. In conclusion, the *Mitsuokella*, *Lactobacillus*, and *Weissella* genera are the primary genera responsible for the formation of kimchi’s flavor.

This study focused on refrigeration-induced quality changes in aquaculture-derived large yellow croaker at 4 °C. Additionally, it examined alterations in bacterial composition as well as microbial diversity during refrigeration using 16S rDNA sequencing. Furthermore, the dominant spoilage bacteria present in large yellow croaker were isolated and identified at the end of the storage period, while their decay-causing ability was quantitatively measured. The aim of this study is to provide a theoretical reference for the future development of preservation techniques aiming to extend the shelf life of large yellow croaker.

## 2. Materials and Methods

### 2.1. Sample Treatment

Live large yellow croaker (body length: 30 ± 2 c; mass: 350 ± 30 g) from a large cage culture in Ningde, Fujian Province, were placed in crushed ice and sent to our laboratory within half an hour of being caught. Then, the scales, heads, tails, and internal organs were removed. The fish were rinsed with water and drained before being placed in 200 mm × 300 mm self-sealing bags and stored in a refrigerator at 4 °C. For the determination of total viable count (TVC), total volatile basic nitrogen (TVB-N), and thiobarbituric acid-reactive substances (TBARS), the fish meat was minced at 0, 2, 4, 6, 9, 12, and 15 days. For 16S rDNA sequencing, the minced fish meat was collected at 0, 3, 6, 9, 12, and 15 days. The samples were designated as 0d, 3d, 6d, 9d, 12d, and 15d, with a minimum of three parallel replicates for each group.

### 2.2. Determination of TVC

According to GB 4789.2-2022 “National Food Safety Standard—Microbiological Examination of Foods—Determination of Total Viable Count” [17], 25 g of sample was homogenized in 225 mL of sterile physiological saline in a homogenizing bag. A 1:10 dilution sample homogenate and prepared serial dilutions with a 10-fold dilution factor were prepared. Two appropriate dilutions of the sample homogenate were selected, and 1 mL of each dilution was pipetted onto a total-viable-count agar plate. The results were counted after culture at 30 °C for 48 h and calculated according to the standard.

### 2.3. Determination of TVB-N

Following GB 5009.228-2016 “National Food Safety Standard—Determination of Total Volatile Basic Nitrogen in Foods” [18], 20 g of sample was placed in a stoppered conical flask, and 100 mL of distilled water was accurately added. The mixture was soaked for 30 min and filtered, and then, the filtrate was collected for testing. If the filtrate could not be tested immediately, it was stored at 4 °C.

### 2.4. Determination of TBARS

Referring to the method by Lan [19] with modifications, 5 g of sample was accurately weighed and added to 25 mL of 20% trichloroacetic acid (Sinopod Chemical Reagent Co., LTD., Shanghai, China). The mixture was shaken for 1 min, and then, centrifuged at 4 °C and 8000 rpm for 10 min. The entire supernatant was brought up to a final volume of 50 mL. Then, 5 mL of the solution was added to 5 mL of 0.02 M TBA solution. The mixture was heated in a water bath at 90 °C for 40 min, and then, cooled to room temperature. The absorbance was measured at 532 nm.

### 2.5. 16S rDNA Sequencing

The samples were ground to a powder via liquid nitrogen grinding, and DNA was extracted using an E.Z.N.A™ Mag-Bind Soil DNA Kit (OMEGA Bio-tek, Guangzhou, China). The integrity of the DNA was verified via agarose gel electrophoresis, and the concentration of the DNA samples was quantified using Qubit. After two rounds of PCR amplification, a library was constructed and sequenced.

Referring to the method by Zhang et al. [20], the traditional plate separation method was used. The fish meat samples from day 15 were homogenized in sterile water and serially diluted. Then, 50 μL of different dilutions of bacterial suspension were spread onto LB agar plates and *Pseudomonas* CFC selective medium plates, respectively, and incubated at 30 °C for 48 h. Single bacterial colonies with different morphologies were picked and streaked three times for purification until pure single colonies were obtained. The purified strains were cultured overnight in LB liquid medium, and genomic DNA was extracted using an SK8255 Ezup Column Bacterial Genomic DNA Extraction Kit. The extracted DNA was subjected to PCR amplification, and the amplified products were sent for sequencing at Shanghai Sangon Biotech Co., LTD. (Shanghai, China). The remaining pure bacterial cultures were preserved with 20% glycerol at −80 °C.

### 2.6. Spoilage Ability Determination

Based on the method of Chen et al. [21], aseptic fish blocks were prepared from large yellow croaker. In the preliminary study, three dominant spoilage bacteria were isolated from large yellow croaker in the late stage of cold storage: *P. psychrophila*, *P. lundensis*, and *P. fluorescens*. These strains were activated and inoculated on *Pseudomonas* CFC selective medium plates, followed by incubation at 30 °C for 48 h. Typical single colonies were picked and inoculated in LB liquid medium, and the cultures were shaken at 30 °C until the bacterial concentration reached 10^6^ CFU/g. Aseptic fish blocks were immersed in suspensions of *P. psychrophila*, *P. lundensis*, and *P. fluorescens* on a clean bench. After 30 s, the fish meat was removed and drained, and then, packed in sterile self-sealing bags for storage at 4 °C. Samples were taken on days 0 and 9 for total viable count and TVB-N determination.

### 2.7. Data Processing and Analysis

Data processing was performed using SPSS 27.0.1 software. A graphical representation was created using Origin 2022. OTU clustering of the sequencing reads was conducted using Usearch 11.0.667 software. Mothur 1.43.0 was used to calculate α-diversity indices and perform rarefaction analysis. β-diversity data analysis and visualization were conducted using R software 3.6.0. Construction of the phylogenetic tree was performed using Mega-X. All experimental data are presented as the average of three repeated operations.

## 3. Results and Discussion

### 3.1. TVC Value

The total viable count is one of the most important indicators for determining the freshness of aquatic products, reflecting the change in the number of microorganisms during storage, which is the primary factor leading to the spoilage of aquatic products. For large yellow croaker, microorganisms exist in the epidermis, gills, internal organs, and other places in the fish body; after the death of the fish, these microorganisms attached to the fish body begin to grow and reproduce, consuming the nutrients in the fish, and produce secondary metabolites, resulting in deterioration in the quality of the large yellow croaker until it becomes inedible. As shown in Figure 1, the total viable count of fresh large yellow croaker was 3.17 lg/(CFU/g), and with the extension of storage time, TVC showed an “S”-shaped increasing trend, reaching 7.43 lg/(CFU/g) on the 9th day, which exceeded the maximum acceptable level of the total viable count in marine fish as stipulated by the International Commission on Microbiological Specifications for Foods [22].

The early stage of storage may be due to a more complex bacterial phase when the strains are antagonistic to each other, resulting in a slower rise in TVC. In the middle stage of storage, specific spoilage bacteria began to dominate when the microorganisms consumed large amounts of proteins, fats, and carbohydrates in the fish, resulting in a significant increase in the total viable count, and in the late stage of storage, a significant portion of the nutritional content in the fish depleted, resulting in deceleration of the growth rate of the total viable count. This is consistent with the findings of Zhang et al. [23], where the initial bacterial count of fresh large yellow croaker was 3.89 lg/(CFU/g), and the microbial counts showed an increasing trend with the prolongation of storage time, and exceeded the acceptable range on the 15th day. Dong et al. [24] applied fish gelatin films containing lauroyl arginine ethyl ester for the preservation of large yellow croaker fillets and found that the initial bacterial counts were all 3.9 lg/(CFU/g). With the extension of storage time, the control group exceeded the maximum acceptable total bacterial count on the 8th day, while the total viable count for the treatment group was 6.8 lg/(CFU/g), which had not yet exceeded the limit value. The above results indicate that the TVC of large yellow croaker is influenced by the treatment method. The initial bacterial count generally fell within the range of 3.5 to 4.0 lg/(CFU/g), while the use of certain antibacterial agents or film packaging methods prolonged the time when the TVC of large yellow croaker exceeded the standard.

### 3.2. TVB-N Value

Volatile basic nitrogen (TVB-N) is an important index for judging the freshness of aquatic products, and is a salt-based nitrogen-like substance formed by combining alkaline nitrogenous substances generated during the decomposition of proteins in fish with organic acids, and the change in its content can reflect the process of spoilage in aquatic products [25]. According to the standard of SC/T 3101-2010, “Fresh large yellow croaker, frozen large yellow croaker, fresh small yellow croaker, and frozen small yellow croaker”, the TVB-N value of qualified products of large yellow croaker should be ≤30 mg/100 g [26]. As shown in Figure 2, the TVB-N value on the 0th day was 2.45 mg/100 g, and with the prolongation of storage time, the TVB-N showed an increasing trend. It increased to 12.78 mg/100 g on the 9th day, which was close to the threshold value for first-grade products (15 mg/100 g). From day 9, there was a sharp increase, reaching 42.56 mg/100 g on day 15, which was far more than the acceptable limit value of the qualified produce. At this time, the fish meat had a pronounced putrid odor. This is due to the fact that during storage, the proteins in fish are decomposed into amino acids by tissue hydrolytic enzymes and microorganisms, which are subsequently degraded into substances such as ammonia and amines, leading to the production of unfavorable odors in the fish body. In this study, even though the TVB-N value of large yellow croaker on day 9 did not exceed 30 mg/100 g, the fish meat was relatively soft at this time, with a slower rebound of the depression after finger pressure, a noticeable putrid odor, and more mucus on the surface that was turbid, which was organoleptically determined to have reached the end point of spoilage. Mansur et al. [27] found that there was a difference between the TVB-N value and the total viable count in beef. Under aerobic storage conditions, the total viable count exceeded the limit value on the seventh day, but the TVB-N value did not exceed the standard limit, which was the same as the research conclusion of Chen et al. [28]. These studies collectively indicate that the TVB-N value exhibits a certain degree of lag, and relying solely on this parameter is insufficient to accurately determine the shelf life of products.

### 3.3. TBARS Value

Lipid oxidation is an important cause of fish quality deterioration. The fat content of farmed large yellow croaker is higher than that of its wild counterpart, which is hydrolyzed under the action of lipase to produce alcohols, aldehydes, acids, and other substances, resulting in a fishy odor and accelerating the spoilage process of the fish. Lipid oxidation not only leads to the generation of an unpleasant odor in fish, but it also generates toxic secondary metabolites such as malondialdehyde. Thiobarbituric acid can react with malondialdehyde to produce a red-colored complex, and the degree of lipid oxidation can be determined by measuring the absorbance of this complex. The larger the TBARS value, the deeper the degree of lipid oxidation [29]. As depicted in Figure 3, the initial TBARS value of fresh large yellow croaker is 0.46 mg/100 g. With the extension of storage time, the overall trend of TBARS increased, and it was 0.60 mg/100 g on the 15th day. In the middle and late stages of storage, with the increase in the number of microorganisms, the enzymes in the microorganisms accelerated the process of lipid oxidation in the fish, and the trend is in line with the results of the study by Hui et al. [30]. This is also a consistent with the findings of Chai et al. [31], who found that ultrasound and rosemary could synergistically inhibit the activities of endogenous lipase and lipoxygenase associated with oxidation, thereby suppressing the process of lipid oxidation during the storage of large yellow croaker. The shelf life of large yellow croaker can also be extended by employing techniques in physical, chemical, and biological preservation to inhibit lipid oxidation.

### 3.4. OTU Clustering Analysis

In sequencing data, one sequence corresponds to one read, and the valid sequences are the data obtained after quality control of the raw data. The valid sequences were subjected to OTU clustering analysis, and based on the results of the clustering analysis, a variety of diversity index analyses can be analyzed. OTU stands for Operational Taxonomic Unit, which groups sequences together based on their similarity to understand community distribution information in sample sequencing, and a collection of sequences represents an OTU. The bacterial sequencing information of the large yellow croaker samples during the 4 °C storage process is shown in Table 1. A total of 410,300 valid sequences were detected from the flesh of large yellow croaker, generating an average of 68,376 sequences per sample, and the average length of valid sequences was 425 bp. To enhance the efficiency and accuracy of the analysis, the sequenced sequences were clustered into OTUs based on a similarity threshold of 97%, and a total of 1897 OTUs were clustered.

### 3.5. α-Diversity Analysis

The microbial diversity and other microorganisms of large yellow croaker during storage at 4 °C were analyzed using high-throughput sequencing, and the Alpha diversity index is presented in Table 2. The coverage values in Table 2 are all above 0.99, indicating that the results of this sequencing can represent the real situation of the samples. The Chao index and Ace index exhibited a decreasing trend followed by an increasing trend, in which the increase from 12d to 15d may be attributed to changes in the community structure caused by the accumulation of microbial metabolites, which is similar to the research conducted by Situ et al. [32]. The Chao index and Ace index were higher than the observed number of OTUs in the corresponding samples, indicating that there were some additional microbial species that had not yet been detected. A higher Shannon index indicates higher species diversity, while the Simpson index shows the opposite [33]. The overall decreasing trend of the Shannon index and the overall increasing trend of the Simpson index indicated that a decline in species richness and evenness results in a decrease in microbial diversity with prolonged storage time. This is because, in the early stage of storage, the microorganisms require fewer nutrients, and the microorganisms are in an adaptation phase. Therefore, the microbial abundance and diversity were the highest in the flesh of the large yellow croaker in the early period. In the middle and late stages of storage, protein and other large molecules in fish flesh were degraded under the action of enzymes, which provided abundant nutrients for microbial growth and reproduction. Microbes with stronger competitiveness proliferated rapidly and produced metabolites and toxins, which inhibited the growth of other microorganisms and led to a decline in the diversity of the microbial population diversity. During the storage of aquatic products, the diversity of microorganisms usually shows a decreasing trend. Li et al. [34] found that the bacterial diversity of fresh salmon fillets was the highest, and the diversity gradually decreased during the storage process.

A rarefaction curve can evaluate whether the sequencing volume is sufficient to capture all taxonomic groups and indirectly reflect the species richness in the samples. In Figure 4A, the curves of each sample tend to flatten out, and the number of OTUs does not increase with increasing sequencing depth, indicating that the amounts of sequencing data are sufficient and appropriate. The Shannon–Wiener curves depict the microbial diversity of each sample at different sequencing depths. In Figure 4B, the curves tend to be flat, indicating that the sequencing data are sufficient to respond to the majority of the microbial information in the samples. Furthermore, the 0d sample has the highest number of OTUs, indicating that the sample at 0d had the highest species diversity and the most complex microbial community structure. The sequencing results from Zhang et al. [35] also showed that all the dilution curves tended to flatten out during the storage period of grass carp.

Rank abundance curves are used to explain the richness of species and the degree of species’ evenness in a sample. The length of the curve on the *x*-axis reflects the species richness, with wider curves indicating a more diverse composition of species. The shape of the curve reflects the evenness of the species composition, with flatter curves indicating higher evenness. Based on Figure 4C, it can be observed that the curves at 0d and 3d decline smoothly when the species composition is the richest and the diversity is the highest. From day 9 onwards, the curve sharply decreases, suggesting that the proportion of dominant bacterial taxa in the samples was higher at this time and the diversity was lower, which aligns with the conclusions in Table 2.

### 3.6. β-Diversity Analysis

The distance-based heatmap of the samples displays the similarity level of microbial communities among samples, with different colors representing different groups. The more similar the samples are, the closer they will cluster together on a branch. The color gradient represents the distance measurement, and the bluer the color, the closer the distance between samples. The tree diagram in Figure 5 depicts the hierarchical relationships among samples based on the Bray–Curtis algorithm at the genus level. Samples from 0d and 3d cluster together on one branch, displaying a consistent shade of blue in the corresponding color blocks. Similarly, the samples from 9d, 12d, and 15d also form a distinct cluster, characterized by varying shades of blue. This indicates that the microbial community structures in 0d and 3d exhibit similarity, while those at 9d, 12d, and 15d demonstrate a resemblance, as well. Notably, the sample at 6d initially clusters independently before eventually merging with the samples from 9d, 12d, and 15d. Combined with the analysis of quality changes during the storage process, the TVC and TVB-N show a slow rise from day 0 to 3, remaining at a low level. On day 6, both indicators show a faster rise, and from day 9 to 15, they remain at a high level, exceeding the specified limit. Based on this, the entire storage period can be divided into the early storage period (0d and 3d), the middle storage period (6d), and the late storage period (9d, 12d, and 15d).

In order to compare the changes in microorganisms in the fish flesh of large yellow croaker on different storage days, genus-level Principal Component Analysis (PCA) was conducted. PCA is an analytical method used for data dimensionality reduction that maps data from high-dimensional space to low-dimensional space through linear or non-linear mapping functions. It condenses multiple indicators into a smaller set, effectively reducing redundant information while preserving the essential structure of the data. As illustrated in Figure 6, the PC1 value is recorded as 74.36%, indicating that the variation along the *x*-axis accounts for 74.36% of the total analysis results. The PC2 value stands at 16.13%, and together, PC1 and PC2 explain 90.49% of the overall results, suggesting that the sample information is sufficiently represented [16]. The microbial communities at the beginning and end of the storage period exhibit significant differences, as evidenced by the clustering of data points at 0d and 3d and another cluster at 9d, 12d, and 15d. This clustering implies a higher similarity in bacterial community compositions between these respective periods.

Non-metric Multidimensional Scaling (NMDS) is also a kind of data dimensionality reduction that locates the research objects in a multi-dimensional space and moves them to a low-dimensional space for classification and localization, while keeping the original relationship between the objects unchanged and reflecting the degree of difference in species information between samples using the distance between points. In NMDS analysis, the stress value serves as a measure of the strength of the analysis results. Typically, stress values below 0.2 are considered to have interpretative significance; values below 0.1 indicate good sorting; and values below 0.05 suggest excellent representativeness. Figure 7 illustrates the NMDS analysis with a stress value of 0.02769, indicating that the analysis is highly representative and effectively explains the intergroup differences in the microorganisms present in the flesh of large yellow croaker during different storage periods. At the genus level, there is a similar group structure observed between 0d and 3d, as well as between 9d, 12d, and 15d. This suggests significant differences in microbial community structures between the beginning and end of the storage period. Furthermore, it indicates that toward the end of the storage period, after the dominant species have established themselves, there is little overall change in the group structure. These findings align with the earlier PCA results.

### 3.7. Species Composition

A Venn diagram presents a visual representation of a dataset, employing logic, simplicity, scientific principles, and comprehensiveness to classify its data members. Among the different types of Venn diagrams, the petal diagram stands out. It employs various colors to denote distinct groupings, with the central area indicating the number of samples shared among multiple groups and the data displayed on the petals representing the number of samples unique to each group. As depicted in Figure 8, the Venn diagram provides a count of the overlapping and distinct Operational Taxonomic Units (OTUs) across the groups. In bacterial 16S sequencing, an OTU is generally considered to belong to a genus of microorganisms. Specifically, the 0d, 3d, 6d, and 15d samples exhibited 37 shared OTUs, along with 467, 389, 22, and 42 unique OTUs, respectively. Notably, the highest number of unique OTUs was observed at 0d, suggesting that this time point had the highest species richness and the most intricate diversity. Additionally, both 9d and 12d had zero OTUs unique to them, indicating that the composition of OTUs in these samples was identical during this period. Furthermore, there was a gradual decrease in the number of unique OTUs from 0d to 12d, implying a decline in species diversity. However, the increase in unique OTUs at 15d might be attributed to microbial metabolites causing changes in the community structure, aligning with the findings presented in Table 2.

### 3.8. Relative Abundance of Bacteria

Bubble plots are effective tools for visualizing the differences in the fractional abundance of species between samples. In Figure 9, the relative abundance of bacteria at the phylum level in large yellow croaker is presented. The microbial composition in fresh large yellow croaker primarily consisted of microorganisms from the phyla Proteobacteria, Firmicutes, and Bacteroidetes, accounting for 36.70%, 20.28%, and 15.67%, respectively. As the storage time increased, the proportion occupied by Firmicutes gradually increased. By 9d, it reached 95.38% and remained above 95% thereafter, making Firmicutes the dominant bacterial phylum in 4 °C refrigerated large yellow croaker. Conversely, the proportion of Firmicutes decreased gradually from 20.28% to 2.39%.

Moving on to Figure 10A, at the class level, the bacterial flora in fresh large yellow croaker mainly consisted of Gammaproteobacteria, Clostridia, Alphaproteobacteria, Betaproteobacteria, and others. Among them, Gammaproteobacteria, Alphaproteobacteria, and Betaproteobacteria belongs to the phylum Proteobacteria, while Clostridia belongs to the phylum Firmicutes. The proportional changes observed in each class were consistent with those observed at the phylum level. At the order level (Figure 10B), the dominant orders in large yellow croaker include Pseudomonadales, Alteromonadales, Aeromonadales, Bacillales, Flavobacteriales, and others.

As depicted in Figure 10C, the dominant families identified in large yellow croaker were Pseudomonadaceae, Moraxellaceae, Aeromonadaceae, Shewanellaceae, Flavobacteriaceae, Listeriaceae, and others. The genus composition of fresh large yellow croaker is characterized by complexity, with a balanced proportion observed for each genus (Figure 10D). By 6d, *Aeromonas*, *Acinetobacter*, *Pseudomonas*, *Brochothrix*, and *Shewanella* began to dominate, accounting for 42.05%, 28.99%, 12.40%, 6.55%, 3.5%, 3.5%, 3.5%, and 4.5%, respectively. Notably, the proportion of *Pseudomonas* spp. gradually increased with prolonged refrigeration time, reaching 82.33% at 15d, thus becoming the predominant genus in large yellow croaker.

*Pseudomonas*, belonging to the family Pseudomonadaceae, is a group of cold-tolerant aerobic Gram-negative bacteria with robust metabolic and spoilage capabilities. Additionally, they can inhibit the growth of other microorganisms through siderophores [36,37]. The *Pseudomonas* genus encompasses species such as *P. aeruginosa*, *P. fragi*, *P. lundensis*, *P. fluorescens*, *P. putida*, and *P. azotrophicus*. In aquatic products, common *Pseudomonas* species include *P. fluorescens*, *P. putida*, and *P. azotrophicus*, among others. Yan et al. [38] isolated *P. fluorescens* from large yellow croaker and turbot, which exhibited strong swimming ability and ferritin-producing activity. Guo et al. [39] isolated *Shewanella putrefaciens* and *P. fragi* at the end of the shelf life of large yellow croaker, with *P. fragi* displaying the most significant activity at 25 °C. Bassey et al. [40] explored the effect of phenyllactic acid on *P. lundensis* and *Brochothrix thermosphacta*, and found that phenyllactic acid played an antibacterial role by destroying the cell walls and causing the leakage of intracellular components, resulting in the death of bacteria. Similarly, Chen et al. [21] found that D-tryptophan inhibited the growth and decay potential of *P. fluorescens* and *Shewanell baltica*. The application of D-tryptophan to the preservation of salmon fillets could extend their shelf life for two days, indicating that exogenous D-tryptophan could extend the shelf life of ice-stored seafood products.

*Aeromonas*, belonging to the family Aeromonadaceae, is a class of Gram-negative bacteria equipped with polar flagella. The genus *Aeromonas* includes over 30 species and subspecies, such as *Aeromonas hydrophila*, *Aeromonas veronii*, *Aeromonas salmonicida*, and *Aeromonas sobria*. It possesses strong environmental adaptability and is widely distributed in nature. *Aeromonas* not only produces aerolysin, but also generates hemolysin, enterotoxin, and other toxins, making it a major pathogenic bacterium in aquaculture. Some studies have demonstrated the significant inhibitory effect of eugenol and chili pepper extracts on the growth of *Aeromonas hydrophila* [41]. Lee et al. [42] found that a salt content of 1.8% was capable of inhibiting the growth of *Aeromonas* in salmonids.

As the storage time increased, the genus structure of large yellow croaker exhibited a decrease from 30 to 21 species, indicating a gradual decline in species diversity. The proportions of *Aeromonas* spp., *Acinetobacter* spp., *Shewanella* spp., and *Brochothrix* spp. displayed an increasing trend followed by a decrease, while the proportion of *Pseudomonas* spp. demonstrated a steady increase. This could be attributed to the stronger adaptive ability and competitiveness of *Pseudomonas* spp. Li et al. [43] discovered that the predominant spoilage bacteria in large yellow croaker during low-temperature storage were *Shewanella* spp. and *Pseudomonas* spp. Zhu et al. [44] found that *Shewanella* spp. (65.12%) was the dominant genus in large yellow croaker stored at 5 °C. The difference between their results and the present study may be attributed to various factors, such as the species of the product [45,46], storage conditions and handling procedures [47,48,49], fishing location, and month [50,51,52]. Moreover, the ability of *Pseudomonas* to replace *Shewanella* as the dominant genus is closely related to its growth and reproduction rates, as well as its group-sensing mechanism.

### 3.9. Species Identification

#### 3.9.1. PCR Amplification and Species Identification Results

At the conclusion of storage, a total of six distinct strains were screened and isolated from fish meat. Among them, three strains were obtained from the *Pseudomonas* CFC selective medium, designated as C-1, C-2, and C-3, while the remaining three strains originated from the LB medium, known as LB-1, LB-2, and LB-3. The 16S rDNA sequences of the six bacterial strains that were screened were compared to the rDNA/ITS database in NCBI using Nucleotide Blast. This comparison aimed to determine the species represented by these strains. Sequences with more than 99% homology were selected and used to construct a phylogenetic tree using Mega X software. The results of the analysis (Figure 11) revealed that C-1, C-2, and C-3 showed similarity to *P. lundensis*, *P. psychrophila*, and *P. fluorescens*, respectively. On the other hand, LB-1, LB-2, and LB-3 exhibited the highest similarity to *Aeromonas salmonicida*, *Aeromonas veronii*, and *Brochothrix thermosphacta*, respectively. It is worth noting that the above-mentioned six species belong to the genera *Pseudomonas*, *Aeromonas*, and *Brochothrix*, respectively. These findings are consistent with the results obtained from the assessment of the relative abundance of bacteria at the genus level.

*P. lundensis* is a bacterium known for causing spoilage in various aquatic products [53,54], refrigerated meat [55], and raw milk [56]. It has the ability to form bacterial biofilms, which help it withstand adverse environmental conditions and promote its proliferation, ultimately leading to product spoilage. *P. fluorescens* is another common spoilage bacterium found in refrigerated aquatic products, meat, and other items. It produces fatty acid enzymes that accelerate lipid oxidation and rancidity in the products. Additionally, it influences the growth and reproduction of other species through siderophore production and the formation of biofilms, thereby maintaining its own population density [57,58]. Research conducted by Jia et al. [59] demonstrated that *P. psychrophila*, as a dominant spoilage bacterium in chub (a type of fish), is capable of causing protein degradation and producing volatile saline nitrogen, putrescine, and strong ammonia, all of which contribute to product spoilage. Yuan et al. [60] also isolated *P. psychrophila* from raw milk, where it acted as the third-most prevalent species within the *Pseudomonas* genus. In conjunction with *P. fluorescens* and *P. fragi*, it contributed to the deterioration of raw milk quality.

#### 3.9.2. Quantitative Analysis of Corruption-Causing Capacity

In order to compare the spoilage abilities of the three isolated *Pseudomonas* species during the later stages of storage, the production factors of spoilage metabolites were taken into consideration. The production factor value indicates the quantity of spoilage metabolites produced by a unit of spoilage bacteria at the end of the product’s shelf life. A higher production factor value indicates a stronger spoilage ability of the bacterium [61]. According to the results presented in Table 3, the spoilage abilities of the three species are ranked from strongest to weakest as follows: *P. flurescens*, *P. lundensis*, and *P. psychrophila.*

## 4. Conclusions

During the refrigeration process, large yellow croaker undergoes spoilage and deterioration due to the oxidation of lipids, degradation of proteins, and growth and reproduction of microorganisms, resulting in the loss of its original nutritional value. The conclusions drawn from the detection of TVC, TVB-N, and TBARS during the refrigeration process are as follows: With prolonged storage time, the TVC, TVB-N, and TBARS of large yellow croaker show an increasing trend, and the rise in these three factors is significantly correlated with storage time. Within a certain range, higher TVC values correspond to higher TVB-N and TBARS values. Through 16S rDNA sequencing, it was discovered that with prolonged storage time, species richness, species evenness, and species diversity all show a decreasing trend, which is related to the dominance of dominant spoilage bacteria in the later stages of storage. Further research on the spoilage characteristics of dominant spoilage bacteria and the development of effective preservation techniques targeting these bacteria will provide a theoretical basis for extending the shelf life of large yellow croaker and other marine products.

## Figures and Tables

**Figure 1 foods-12-03994-f001:**
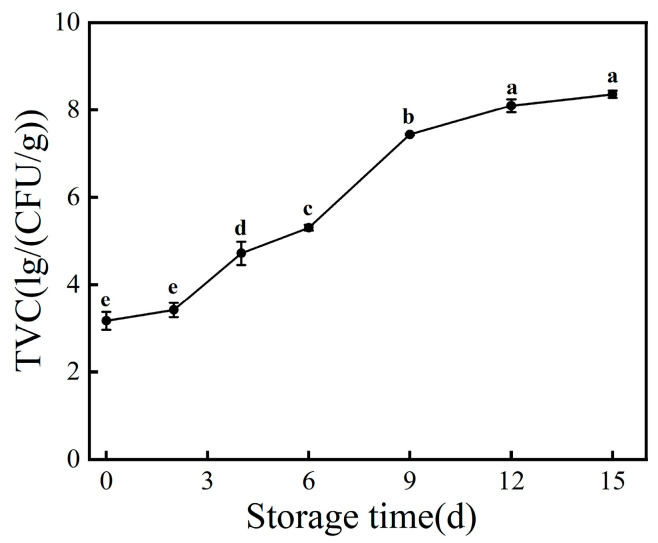
Changes in TVC of large yellow croaker during storage at 4 °C. Each data point is the mean of three replicates. Different letters represent significant differences (*p* < 0.05).

**Figure 2 foods-12-03994-f002:**
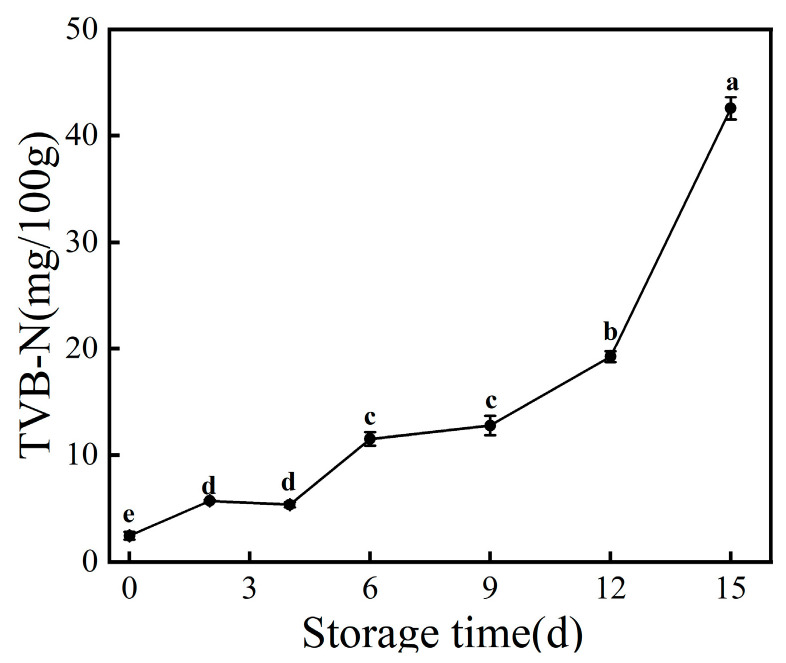
Changes in TVB-N values of large yellow croaker during storage at 4 °C. Each data point is the mean of three replicates. Different letters represent significant differences (*p* < 0.05).

**Figure 3 foods-12-03994-f003:**
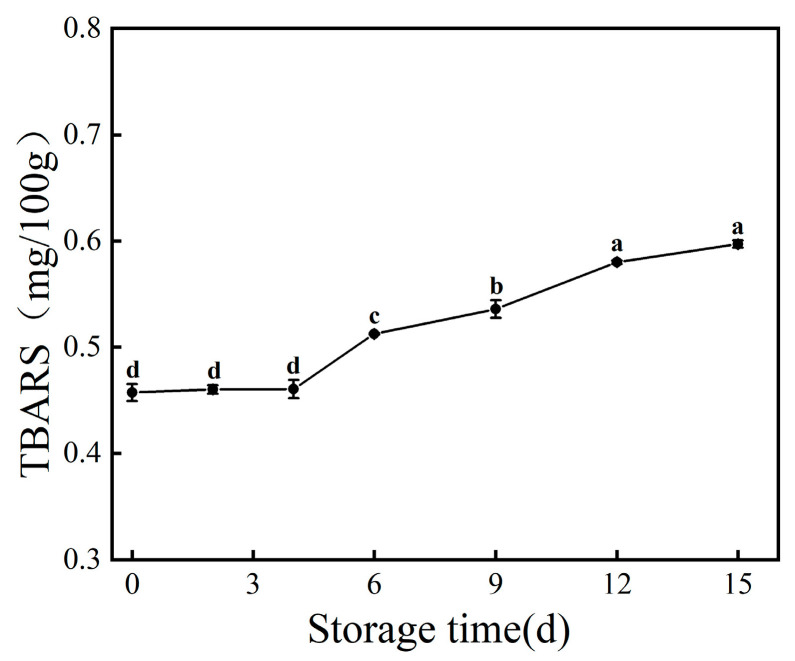
Changes in TBARS values of large yellow croaker during storage at 4 °C. Each data point is the mean of three replicates. Different letters represent significant differences (*p* < 0.05).

**Figure 4 foods-12-03994-f004:**
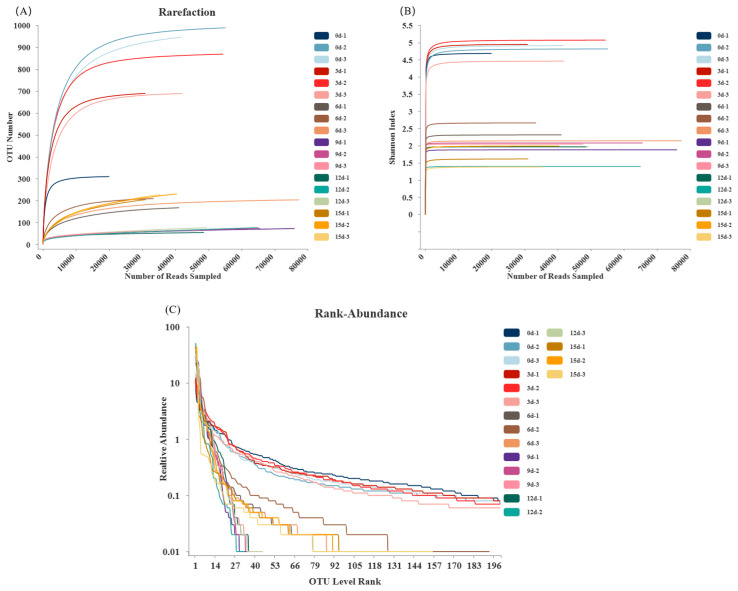
16S rDNA sequencing results of large yellow croaker during storage at 4 °C. (**A**) Rarefaction curve; (**B**) Shannon–Wiener curve; (**C**) Rank abundance curve.

**Figure 5 foods-12-03994-f005:**
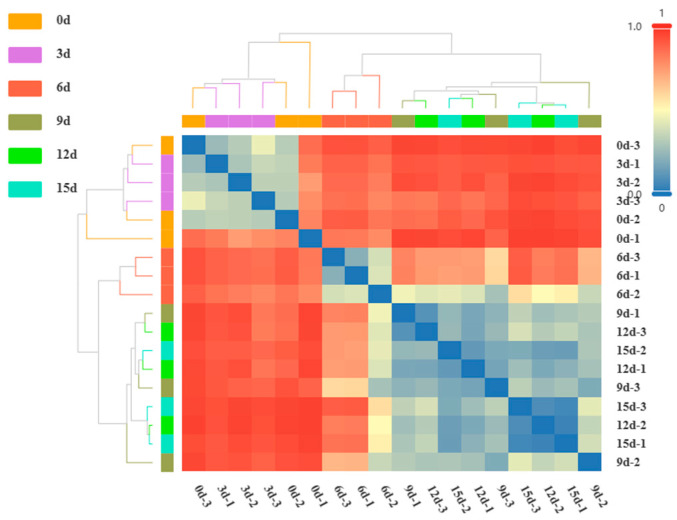
Distance heatmap of large yellow croaker during storage. Blue means the samples are close together; red means the samples are far apart. The bluer the color, the closer the distance. The redder the color, the farther the distance.

**Figure 6 foods-12-03994-f006:**
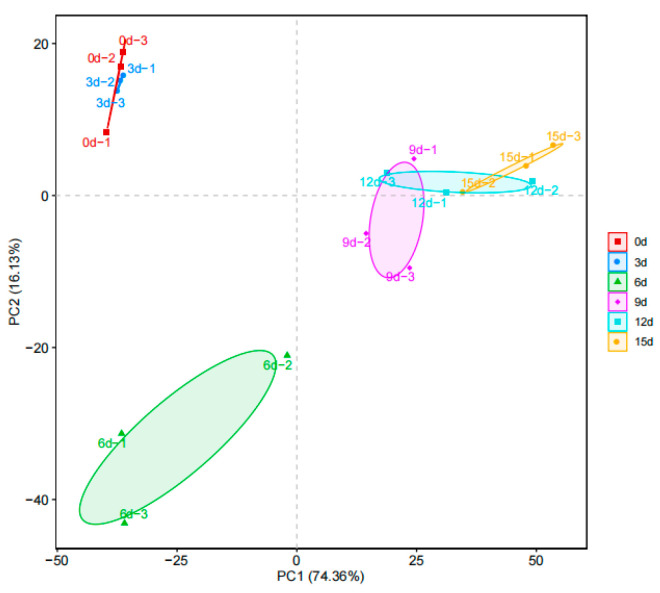
Beta diversity comparison of the microbial diversity of large yellow croaker during storage as per Bray–Curtis dissimilarities matrix, represented by Principal Component Analysis (PCA).

**Figure 7 foods-12-03994-f007:**
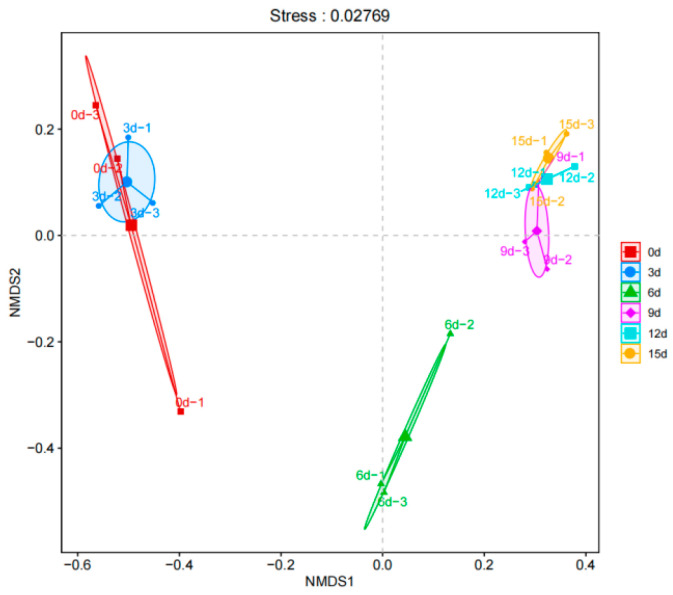
Beta diversity comparison of the microbial diversity of large yellow croaker during storage represented by Non-metric Multidimensional Scaling (NMDS).

**Figure 8 foods-12-03994-f008:**
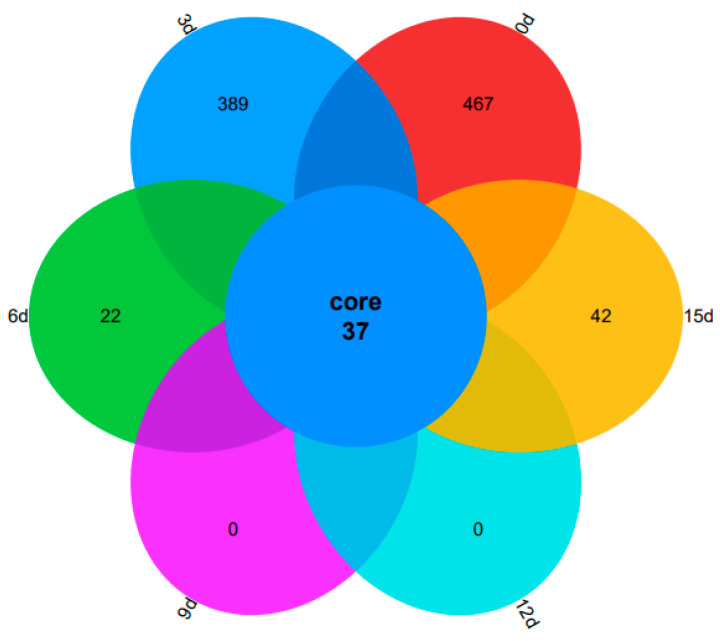
Venn diagrams of the unique and common OTUs of the microbial communities in large yellow croaker.

**Figure 9 foods-12-03994-f009:**
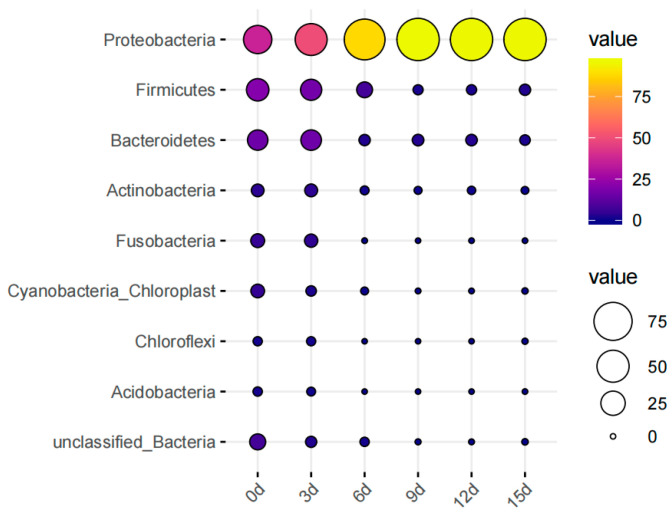
The bacterial community composition at the phylum level during storage of large yellow croaker.

**Figure 10 foods-12-03994-f010:**
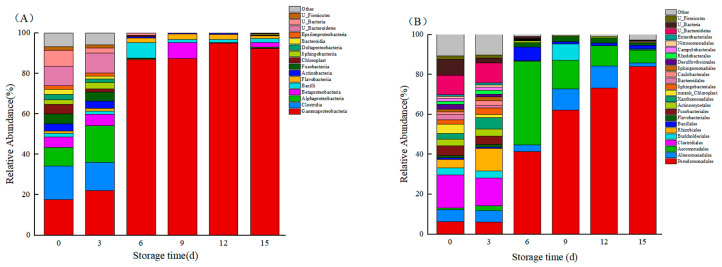
Microbial community composition of large yellow croaker during storage. (**A**) Relative abundance of bacteria in large yellow croaker at the class level. (**B**) Relative abundance of bacteria in large yellow croaker at the order level. (**C**) Relative abundance of bacteria in large yellow croaker at the family level. (**D**) Relative abundance of bacteria in large yellow croaker at the genus level.

**Figure 11 foods-12-03994-f011:**
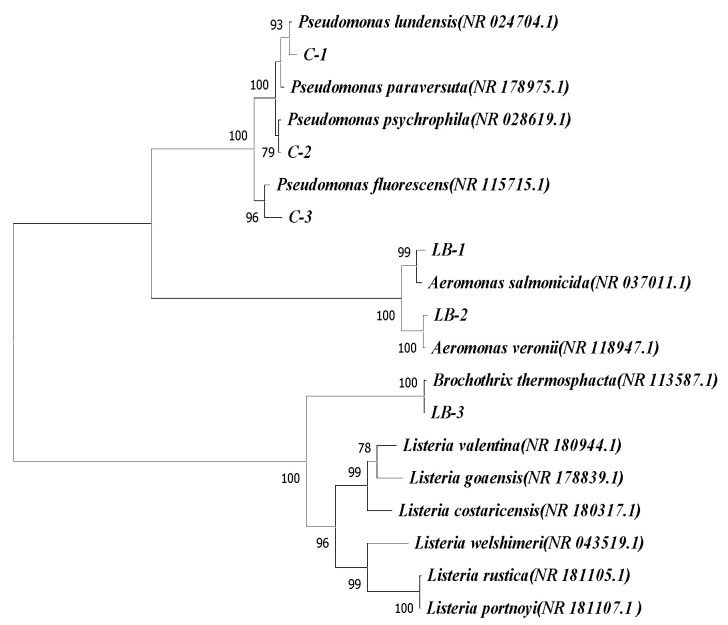
Neighbor-joining phylogenetic tree based on 16S rDNA gene sequence. All bootstrap values are greater than 70% of 1000 bootstraps.

**Table 1 foods-12-03994-t001:** Statistics of valid sequence data of large yellow croaker during storage.

Sample	Number of Effective Sequences (Pieces)	Base Number	Average Length (bp)	Minimum Sequence Length (bp)	Maximum Sequence Length (bp)
0d	65,028	26,949,661	413.22	350	476
3d	66,689	27,712,508	415.40	350	474
6d	62,350	26,678,327	427.74	350	476
9d	87,343	37,428,755	428.53	350	473
12d	79,654	34,139,914	428.60	350	475
15d	49,190	21,063,294	428.21	351	470

Under the Samples column, 0d, 3d, 6d, 9d, 12d, and 15d represent large yellow croaker stored for 0, 3, 6, 9, 12, and 15 days, respectively.

**Table 2 foods-12-03994-t002:** Alpha diversity of microorganisms in large yellow croaker during storage.

	Number	OTUs	Chao	Ace	Shannon	Simpson	Shannoneven	Coverage
0d	38,585	719	732.444 ^a^	728.687 ^a^	4.717 ^a^	0.031 ^d^	0.739	0.99924
3d	41,710	717	722.957 ^a^	720.287 ^a^	4.783 ^a^	0.030 ^d^	0.728	0.99955
6d	50,503	172	179.692 ^b^	179.430 ^b^	2.357 ^b^	0.168 ^c^	0.458	0.99966
9d	63,184	71	84.547 ^b^	105.013 ^b^	2.010 ^bc^	0.212 ^bc^	0.471	0.99972
12d	54,360	69	108.942 ^b^	128.171 ^b^	1.781 ^c^	0.283 ^ab^	0.423	0.99962
15d	35,676	225	294.371 ^b^	287.186 ^b^	1.670 ^c^	0.334 ^a^	0.308	0.99815

Values within a column with different superscripts are significantly different (*p* < 0.05).

**Table 3 foods-12-03994-t003:** Results of corruption-causing capacity of *P. psychrophila*, *P. lundensis*, and *P. flurescens*.

Putrefying Bacteria	TVC/CFU		TVB-N (mg/100 g)		γ_TVB-N/TVC_ (10^−2^ mg TVB-N/CFU)
	Initial Stage/10^4^	Last Stage/10^8^	Initial Stage	Last Stage	
*Pseudomonas psychrophila*	5.99 ± 0.37	13.90 ± 0.43	10.03 ± 0.19	20.77 ± 0.83	7.75 × 10^−9^ ± 0.71
*Pseudomonas lundensis*	6.85 ± 0.63	9.47 ± 0.63	9.45 ± 0.34	17.03 ± 0.19	8.08 × 10^−9^ ± 0.43
*Pseudomonas flurescens*	4.23 ± 0.34	10.12 ± 0.20	8.95 ± 0.34	17.73 ± 0.19	8.71 × 10^−9^ ± 0.78

## Data Availability

The data used to support the findings of this study can be made available by the corresponding author upon request.

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
