# Peer review of "Dynamic Changes in the Microbial Composition and Spoilage Characteristics of Refrigerated Large Yellow Croaker (Larimichthys crocea) during Storage"

_foods, 2023, doi:10.3390/foods12213994_

Round 1
Reviewer 1 Report
Comments and Suggestions for Authors
"Dynamic Changes in Microbial Composition and Spoilage Characteristics of Refrigerated Large Yellow Croaker during 4°C Storage" is quite detailed and appears to be part of a rigorous scientific investigation. However, like any document, there are always opportunities for improvement. Here's a breakdown:
The abstract, while comprehensive, is lengthy. Abstracts should be concise yet informative. Consider streamlining sentences and eliminating redundancy. The mention of Pseudomonas being a dominant genus in refrigerated large yellow croaker is repeated. Combine these details or present them once. Begin the introduction with a more direct statement about the topic's significance. There are jumps from the general significance of the fish to very detailed specifics about preservation methods. Consider smoothing these transitions. Some information, like the intricacies of next-generation sequencing, maybe too detailed for an introduction. Consider summarizing these or shifting them to the methodology. Some methods appear redundant, such as the inclusion of both "Experimental samples and reagents" twice. Ensure that there is no repetition. Some procedures are mentioned in passing, e.g., "DNA was extracted." For a methodology section, it might be useful to elaborate slightly on this, e.g., the method or kit used for DNA extraction. Ensure consistent use of units, abbreviations, and nomenclature across the manuscript. Consider using subheadings for each method or approach to make the section more navigable. Ensure that terms are consistently used. For example, choose between "large yellow croaker" and "yellow croaker" and use it uniformly across the text. The text seems to be very dense with a lot of information. Breaking it down into smaller, more concise paragraphs with subheadings can improve its readability. The text switches between different measurement units. Ensuring consistent usage of units can prevent confusion. Define abbreviations the first time they appear in the text. For instance, "TVC" should be defined before being used throughout the text. While the study often references other works for comparison, a deeper analysis comparing the findings of the current study with those of previous studies can provide a clearer context and significance. There's a mention of the paper strip method and plate counting method. A brief introduction or explanation about these methods might be useful for readers unfamiliar with these terms. The section provides results and comparisons, but recommendations or suggestions based on findings can add value. Consider including potential implications of the findings for industry practices or suggestions for future research in this area. While the storage periods are divided into early, middle, and late, it's not clear why D6 is considered the middle storage period. The rationale behind this specific division should be elaborated upon, considering the microbial dynamics and spoilage indicators. It would be beneficial to include statistical analyses to confirm the significance of observed differences at various storage times. Tests like ANOVA or t-test could provide a clearer idea if the differences between the days are statistically significant.
The study uses NMDS to complement PCA in understanding community composition. A deeper dive into NMDS results, possibly indicating beta diversity measures or specific taxa driving differences, would be insightful. While Venn diagrams are utilized to represent shared and unique Operational Taxonomic Units (OTUs), the functional implications of these shared and unique OTUs aren't discussed. Are these OTUs known spoilage bacteria, or do they have other functional roles in the fish microbiome? The study could further elucidate the specific spoilage metabolites produced by dominant bacteria. Connecting the spoilage bacteria's metabolic capabilities to the actual spoilage profile of the fish would be more informative. The article mentions differences in findings from other studies due to various factors like fishing location and handling procedures. It might be beneficial to detail the specific environmental and handling conditions under which the Large Yellow Croaker was stored in this study. This will help contextualize findings and improve repeatability in future studies. Microbial community dynamics isn't just about individual species' presence but also their interactions. Elaborating on potential synergistic or antagonistic interactions between dominant microbial species would be beneficial. After identifying the major spoilage organisms, the study could hint at potential interventions, such as specific antimicrobials or preservation techniques, to target these organisms and extend the shelf life of the fish. While genus-level identification provides useful insights, strain-level differences can significantly impact spoilage capability. A deeper analysis at the strain level could further refine our understanding of spoilage dynamics.
Comments on the Quality of English LanguageModerate editing of English language required
Reviewer 2 Report
Comments and Suggestions for Authors
The manuscript is very interesting and employs robust techniques to achieve the proposed objectives. The generated information may be of great interest to the entire international scientific community. However, to enhance the visibility of the manuscript, several proposed changes are outlined below.
1. Initially, the introduction provides the necessary background but focuses too much on describing the techniques used. It could be more concise regarding the description of the importance of the 16S sequencing technique.
2. The methodology requires adjustments. Various parts of the methodology are in the results section and need to be relocated (lines 290-316, for example).
3. The results and discussion section needs to be reviewed, as several paragraphs and sentences lack references. Additionally, there is little discussion in the initial results. I believe that presenting the manuscript with separate results and discussion sections would enrich the article's presentation and understanding by readers.
4. All table and figure titles should be improved.
5. There is too much explanation of techniques in the results and discussion section. This should be in the introduction.
6. The conclusion is a repetition of results and needs to be reformulated.
Minor remarks:
· Include the scientific name of the species in the title and at the beginning of the manuscript.
· In the title, use only "storage"; the temperature is not necessary in the title.
· Please include a conclusion in the abstract.
· Remove words already present in the section title of Keywords.
· Lines 54-55 - review the spelling of all scientific names (italicized). "Strain" is spelled incorrectly. Several scientific names are not italicized in the introduction (and perhaps in the manuscript). Double-check it.
· The introduction is too long and descriptive regarding the importance of microbial contamination identification using 16S sequencing. Be concise about this. Shorten all of this context and this extremely long paragraph.
· Is there information on whether the fish included in the study were of extractivist or aquaculture origin? As the authors point out in the introduction, this could influence the results obtained.
· Sections 2.1.1 and 2.1.2 are unnecessary. Reagents and equipment can be described throughout the other sections where they are mentioned.
· Line 217 - were the fish purchased while still alive? It is not clear in the article.
· 2.2.2. - use past tense in the methodology and provide a bibliographic reference.
· 2.2.3 - provide a reference.
· The first paragraph of the results section is unnecessary and out of context for the article.
· Many paragraphs lack references.
· Figure 1 - improve the title. Colonies of what?
· The 16S is confusing and mixed in with the subtopic numbering. Review.
· Figure 4 is unnecessary."
·
Author Response
请参阅附件。

Round 2
Reviewer 1 Report
Comments and Suggestions for Authors
Accept in present form
Reviewer 2 Report
Comments and Suggestions for Authors
The manuscript has been accepted and all my suggestions were done.